# Early detection of internet trolls: Introducing an algorithm based on word pairs / single words multiple repetition ratio

**Sergei Monakhov** *

Friedrich Schiller University, Jena, Germany

* sergei.monakhov@uni-jena.de

**Data Availability Statement:** All files necessary to replicate the study findings are available from the Zenodo database (DOI: 10.5281/zenodo.3935816, URL: https://doi.org/10.5281/zenodo.3935816).

## Abstract

Troll internet messages, especially those posted on Twitter, have recently been recognised as a very powerful weapon in hybrid warfare. Hence, an important task for the academic community is to provide a tool for identifying internet troll accounts as quickly as possible. At the same time, this tool must be highly accurate so that its employment will not violate people's rights and affect the freedom of speech. Though such a task can be effectively fulfilled on purely linguistic grounds, as of yet, very little work has been done that could help to explain the discourse-specific features of this type of writing. In this paper, we suggest a quantitative measure for identifying troll messages which is based on taking into account certain sociolinguistic limitations of troll speech, and discuss two algorithms that both require as few as 50 tweets to establish the true nature of the tweets, whether 'genuine' or 'troll-like'.

## Introduction

In February 2018, the U.S. Justice Department indicted 13 Russian nationals associated with the Internet Research Agency, based in St. Petersburg, for interfering with the 2016 U.S. presidential election [1]. These individuals were accused of using false American personas to operate social media pages and groups designed to attract American audiences and sow discord in the U.S. political system by posting derogatory information about a number of candidates, disparaging Hillary Clinton and supporting the campaign of then-candidate Donald J. Trump.

The repercussions of these events were far-reaching: they almost led to the impeachment of the U.S. president, severely damaged the Russian economy due to the U.S. sanctions imposed, and almost ruined relations between the world's two superpowers. Much more than that, these events epitomised the ongoing changes in modern online communication, giving prominence to such concepts as 'fake news', 'alternative facts', and 'post-truth'.

Taking into account the global scale of this scandal and its ever-widening ramifications for society, one can only wonder why the phenomenon of troll writing has not received, to date, any substantial scientific attention. Although the Russian trolls' tweets were collected and examined, as were many other such tweets before them, to the best of our knowledge they were not analysed as a language phenomenon in its own right.

One could argue that before the aforementioned events, trolling was mainly viewed as online impoliteness or cyberbullying, encouraged by anonymity and undertaken for

**Funding:** The author(s) received no specific funding for this work.

**Competing interests:** The authors have declared that no competing interests exist.

amusement's sake [2–4]. The term itself can be traced back to the 1980s; since then, trolling has been investigated within two general frameworks, namely computer-mediated communication [5–7] and hate speech [8–11]. Even the formation of the term reveals that most researchers were interested in behavioural practices and their psychological underpinnings. Hence, we have 'trolling' as widely used name of the process but find little mention of 'troll language' or 'troll speech' as a label for a specific type of discourse.

When troll messages, primarily tweets, were recently recognised as a powerful weapon in modern hybrid warfare, academic and public attention understandably shifted to extensive data mining to identify some extralinguistic factors grouping suspicious posts together. An array of tweet parameters has come under scrutiny: dates of publication, rates of account creation and deletion, language, geographical location, percentage of tweets including images and videos, hashtags, mentions, and URLs. However, very little work has been done that could help to explain the discourse-specific features of this type of writing (see [12] for comprehensive review).

In addition, in the very rare instances of research conducted in the area of content analysis, the findings have been regrettably trivial. For example, in their paper, Zannettou et al. discovered that Russian trolls tend to 1) post longer tweets, 2) be more negative and subjective, and 3) refer to more general topics than random Twitter users. Similarly unaspiring from a linguistic perspective are the results of some other recent studies of state-sponsored and inauthentic internet messages [13–15].

The general problem of these studies is that all of them compare the distribution of certain linguistic features in two datasets of tweets in which the common denominator is that the messages are in English, and the only known parameter is that one of them definitely consists of troll messages. The rationale behind selecting these linguistic features in each case may be more or less spurious. More importantly, it is clear that as long as we choose different baseline datasets, we will most likely get different results. Thus, in stark contrast to what Zannettou et al. discovered, Lundberg and Laitinen in [16] argued that the troll messages are shorter than their baseline material and resemble more formal registers in that they contain a higher proportion of nouns. As we will see below, the latter observation is more compatible with our own results than the former.

Only very recently, attempts have been made to approach the problem of troll writing from a slightly different, semiotic perspective. Monakhov [17] showed that a number of features inherent in trolls' tweets are grounded in the sociolinguistic limitations of this type of discourse, which, in essence, is an imitation game. A troll wants to achieve their goal without being identified as trying to achieve it. In other words, their language attitudes must be predefined and moulded by a combination of two factors: first, speaking with a purpose; second, trying to mask the purpose of speaking. Monakhov then contended that the orthogonal nature of these factors must necessarily result in the skewed distribution of different language parameters of trolls' messages and showed some very pronounced anomalies in the distribution of topics and associated vocabulary in Russian trolls' tweets.

This view seems to be intuitively clear if we agree that troll writing is characterised by the omnipresence of a target message (or a small cluster of such messages) underlying each and every concrete topic, however great the range. Suppose that a troll has to write a great number of messages using the word *Trump*. It is not possible to simply continuously repeat the same tweet because that will lead to the exposure of the troll. Hence, it is necessary to use the target word in a variety of different contexts, including those where it may seem incongruous to most speakers. This, in turn, has consequences for the target word's lexical compatibility: its distribution markedly increases, its neighbours become more numerous, and the co-occurrence links between it and other words become artificially strengthened.

Monakhov achieved an accuracy score of 91% in classifying tweets as originating from trolls by means of fairly simple neural network modelling, thus showing that identifying troll writing is not difficult provided there is a large corpus of examples. However, the question arises of how many tweets are needed to detect the aforementioned distributional anomalies; in other words, how quickly one can recognise a troll on the internet. We provide evidence that as few as 50 tweets are sufficient to answer this question. One possible limitation of the proposed method is that, apart from troll messages, it was tested only against tweets of public figures.

## Hypothesis

The above premises can be reformulated as follows. In the tweets that we—for the lack of a better term—will refer to as 'genuine', with every new post from the author, the probability of meeting a word that has already been used should increase. In contrast, in troll writing, with every new tweet, the probability of meeting a word that has already been used should continuously decrease or stay constant, at least until a certain cut-off point is reached. This is inevitable, provided that the initial hypothesis of trolls being forced to use target words in a variety of different contexts is justified.

Our first task was to ascertain the correctness of using this hypothesised parameter, number of repeated words. To achieve this, we obtained the same two datasets that Monakhov worked with: 1) the collection of tweets connected to the Internet Research Agency, a Russian government-owned troll factory, which was made publicly available on Github by Linvill and Warren [18] at https://github.com/fivethirtyeight/russian-troll-tweets/; 2) the collection of daily tweets from both houses of the U.S. Congress posted by Alex Litel on Github under permissive MIT licence at https://github.com/alexlitel/congresstweets/. The data cleaning and preprocessing was also conducted along the lines described in [17]. After all the preliminary work had been done, 1,361,708 troll tweets and 1,223,340 congress tweets remained in the two samples. Given the nature of our datasets, we were not concerned about the presence of so-called social bots or 'augmented humans' [19–21] and did not implement any procedure of finding and filtering out automatically-generated tweets.

From the preprocessed samples, we randomly subsampled 1,000 congress tweets and 1,000 troll tweets, each tweet including no more than 10 stemmed content words because, in our data, congress tweets are, on average, longer than those written by trolls. Then, we chose the first 100 tweets in both subsamples and, using the equation for the binomial distribution, calculated for each $(i+1)$th tweet with the $r$ number of words the probability $p$ of meeting $k$ words that were used at least once in the group of previous tweets containing overall $n$ words. The procedure was repeated 30 times.

$$P(1 \leq k \leq 10) = \sum_{k=1}^{10} \binom{r}{k} p^k (1-p)^{r-k}$$

The averaged results for the tweets from congresspeople and trolls are presented in Fig 1. At first glance, they look disappointing and seem to rebut our hypothesis. The trends are almost identical: the more tweets, the higher probability to meet words that have already been used. However, it is important to keep in mind that we assessed the probability of seeing $k$ words with $k$ ranging from 1 to 10. It goes without saying that trolls should continuously repeat at least some words, their target words, otherwise they will not be able to achieve their secret goals. So a much more interesting question to answer is whether the probabilities of repeating several words in the same tweet are similar to those given in Fig 1.

We constrained $k$ to lie between 3 and 10 and updated the probabilities. The results, averaged across 30 random samples, are visualised in Fig 2. They appear to lend credibility to our

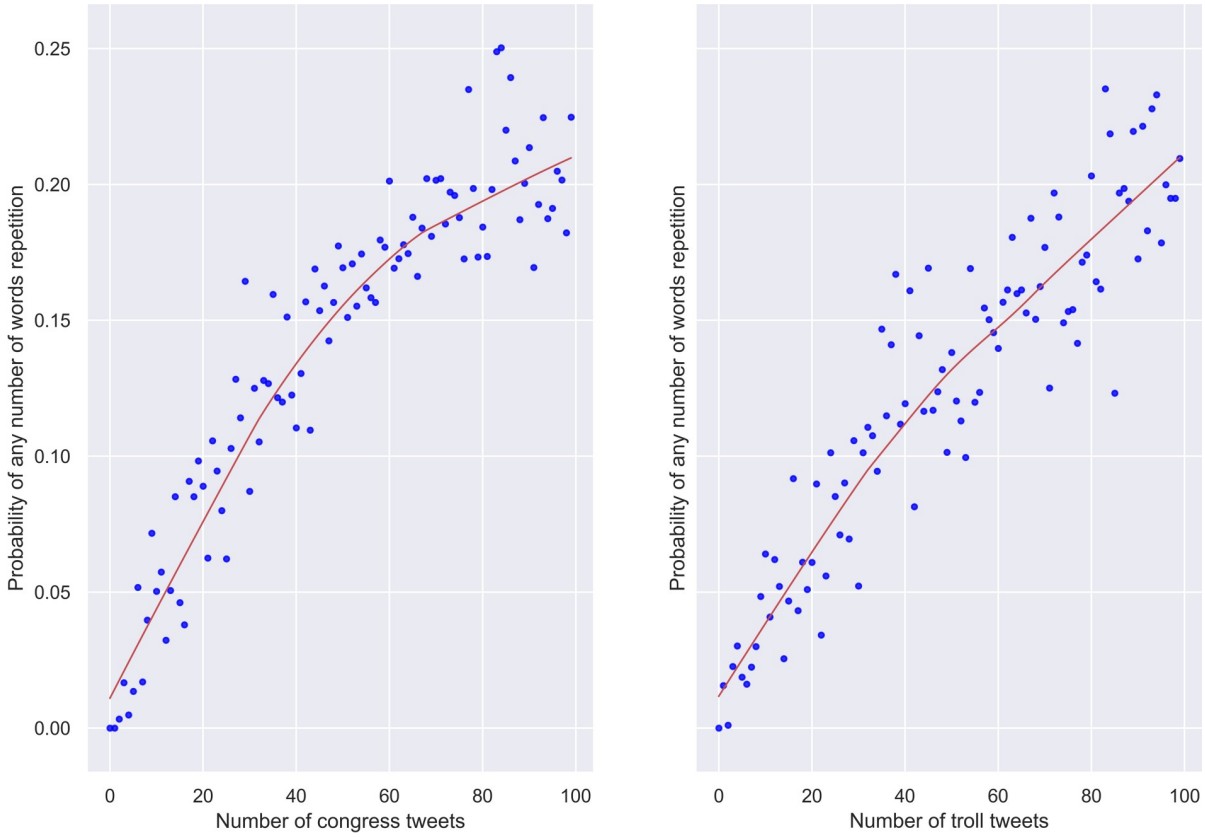

**Fig 1. Congress and troll tweets.** Probabilities of meeting any number of words that have already been used as the number of tweets increases (100 tweets).

speculations since the repetition of 3 and more words at once is much more likely in genuine tweets than in troll tweets.

Finally, in Fig 3 are given the probabilities of seeing any number of repeated words at a stretch of 500 tweets. Interestingly, both types of messaging peak around the same point of 200 tweets but afterwards behave differently. In genuine writing, the probability of meeting a repeated word starts rapidly declining as the number of tweets increases. In contrast, in troll writing, the same probability reaches a plateau and goes down at a much slower rate. Thus, trolls tend to repeat fewer words than congresspeople but do it for a longer time.

However, this parameter on its own cannot be considered sufficient, as it does not prove any malicious behaviour by the author or group of authors. It is possible to imagine a person who, while tweeting, for some reason constantly changes topics and, as a consequence, vocabulary associated with these topics. For this reason, a second parameter is needed, namely the number of repeated pairs of words.

The hypothetical author who simply refrains from saying the same things with the same words has no secret task of delivering a target message multiple times without being suspected of such. They do not need to use a limited number of signal words in a wide variety of different contexts. Therefore, as the number of new words in their tweets increases, so will the number of repeated pairs of words. In contrast, an internet troll who follows the strategy described above will use new words alongside the signal words, which will result in an anomalous distribution of the pairs of words.

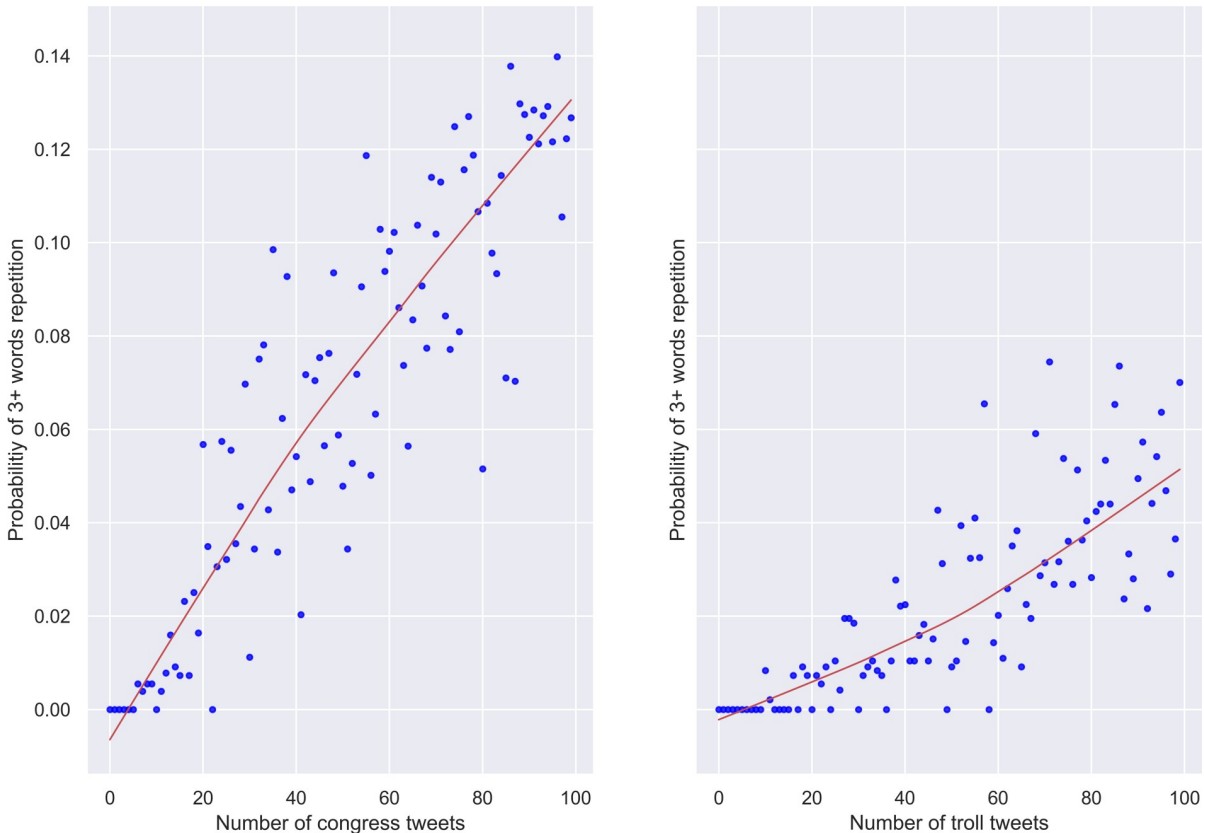

**Fig 2. Congress and troll tweets.** Probabilities of meeting 3+ words that have already been used as the number of tweets increases (100 tweets).

In terms of graph theory, this assumption may be formulated as follows: if we consider all words in a given sample to be nodes and all pairs of words appearing in a tweet to be edges within a network of words from respective tweets, then a clique of words can be defined as a subset of the nodes, such that every two distinct nodes are adjacent (coappear at least in one tweet). We expect to find that the number of maximal cliques, that is, cliques that cannot be extended by including one more adjacent node [22] should be greater in genuine tweets than in troll tweets. This idea is based on the premise of trolls being forced to use target words in a variety of different contexts, which necessarily leads to an increase in the number of open triangles where different filler words may co-appear with target words but not with each other.

To test this assumption, we turned both of our subsamples of 1,000 congress and troll tweets (with length of each tweet ranging from 2 to 10 content words) into networks, in which words constituted nodes and pairs of words appearing in one tweet formed edges [23]. We found a significant association between the type of tweet (genuine or troll) and the distribution of nodes and edges: $\chi^2(1) = 239.97$, $p < 2.2e\text{-}16$. The effect size, however, was marginal, with Cramer's $V$ equal to 0.08. To exclude cases where maximal cliques were formed by the tweets containing hapax legomena, that is, words that do not appear anywhere else in the respective subsample, we made two subgraphs of our networks, taking into account only those nodes and edges that appeared at least four times. The association between the type of tweet (genuine or troll) and the distribution of nodes and edges was also significant: $\chi^2(1) = 26.18$, $p < 2.2e\text{-}16$, and the Cramer's $V$ increased to 0.18. Importantly, the word pairs / single words multiple

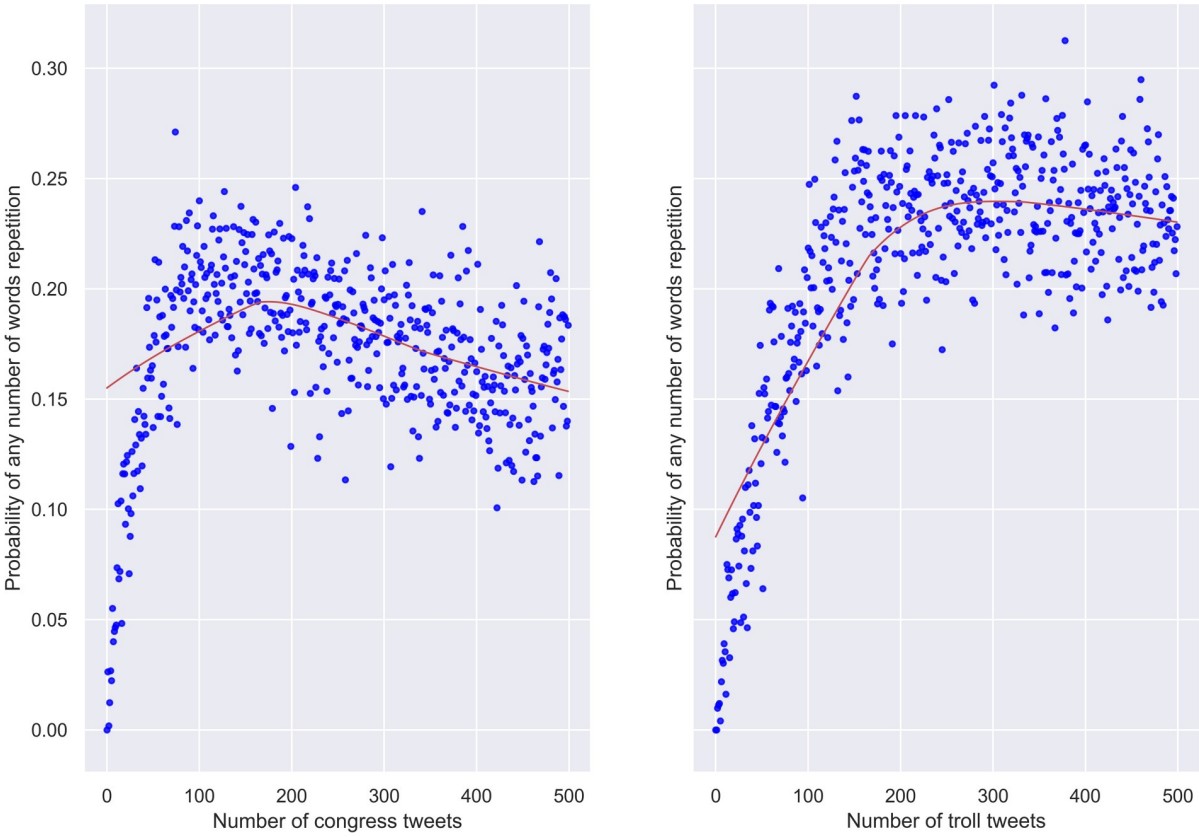

**Fig 3. Congress and troll tweets.** Probabilities of meeting any number of words that have already been used as the number of tweets increases (500 tweets).

repetition ratio in genuine writing is 5.16 times greater than that in troll writing. The visualisation of the respective networks can be found in S1 and S2 Figs.

Much more telling, however, is the distribution of maximal cliques in both subsamples. As we hypothesised, maximal cliques in genuine tweets outnumbered those in troll tweets by a very large margin: 9,760 to 2,069.

Another important observation can be made in this regard. The degree of semantic relatedness of the interconnected words that occur alongside each other at least in two messages is greater in genuine tweets than in troll tweets. To show this, in each subsample we found the node that was a vertex of the largest number of triangles, that is, connected with the largest number of adjacent pairs of words. Then, we combined all thus identified words from the congress and troll tweets into two lists and used the Wordnet [24] to check whether any of these words were a part of the synset of any other word in the respective list. Finally, a simple ratio of the words that were related through their synsets to the total number of words in the list was calculated. For genuine tweets this ratio was found to be 12/49 = 0.24; for troll tweets, 0/37 = 0. This serves as further evidence in favour of the theory that lexical augmentation in troll tweets is driven by the urge to 'dilute' the repeated core message with new words, even though these new words may seem unusual in certain contexts and their appearance there semantically unjustified.

Thus, we can finally formulate our hypothesis as follows. Regardless of the distribution of topics in tweets, we expect to find that correlation of the numbers of repeated content words and content word pairs will be different in genuine and troll writing. In the former, the

number of repeated word pairs will closely follow the number of repeated words. In the latter, however, repeated words will outnumber repeated word pairs.

## Data and methods

To test this hypothesis, we used subsamples of 200 congress and 200 troll tweets as training data to calculate how the ratio of the proportion of repeated content words among all content words to the proportion of repeated content word pairs among all content word pairs changed in genuine and troll writing as the number of accessible tweets increased. The quotient $q$ representing this ratio was calculated separately for groups of different numbers of tweets, ranging from two to 200, and for tweets of different lengths, ranging from two to nine content words:

$$q = \frac{w/W}{p/P}$$

where $w$ and $p$ are the counts of repeated words and word pairs, while $W$ and $P$ are the total numbers of words and pairs in a given group of tweets, respectively. Given our hypothesis, we expected to find that, regardless of the number of content words within a message, tweets written by trolls would be characterised by greater values of $q$ than tweets written by congresspeople. The reason for this is that the denominator should have a higher value in the latter than in the former, since repeated content word pairs are more frequent in genuine writing. After calculating the quotient $q$ for different groups of tweets, we fitted to the data 1) a logistic regression model to analyse whether the difference in $q$ distinguishes between congress and troll tweets significantly; and 2) two linear regression models to predict the values of $q$ for congress and troll tweets with the number of accessible tweets and the number of content words in a tweet as independent variables.

Our next goal was to identify the earliest cut-off point at which tweets of both types can be safely identified. To achieve this, we first combined the values of $q$ obtained for congress and troll tweets into eight pairs of vectors depending on the length of tweet, so that each vector started with the value of $q$ for the group of two tweets and ended with the value of $q$ for the group of 200 tweets: $\{q_2, q_3, q_4 \dots q_{200}\}$. Next, we subtracted the congress vectors from the troll vectors, resulting in eight vectors of differences in the respective values: $\{q_{t2-c2}, q_{t3-c3}, q_{t4-c4} \dots q_{t200-c200}\}$, and then took the mean of each of these eight vectors and identified the indices of the left-most positions equal to the mean. Finally, the median of the obtained indices was taken to establish the optimal number of tweets needed to detect troll writing.

Having achieved this, we combined all previous findings into two algorithms for checking whether a certain group of tweets can be categorised as genuine or troll writing and tested these algorithms on different samples of congress and troll tweets from our collections, as well as on some completely new data. The new data were represented by 32,804 tweets posted by Donald Trump between 2009 and 2017 that we downloaded from https://github.com/mkearney/trumptweets and preprocessed in the same way as the troll and congress tweets.

The tweeting activity of Trump constitutes an interesting case with regard to the nature of the problem that we are discussing. The POTUS is infamous for provocative tweeting; however, unlike the Russian trolls who were engaged in swaying the U.S. elections in his favour, Trump cannot be called a troll in our sense of the word. Specifically, his communicative behaviour is not a combination of two factors: while he undoubtedly tweets with a purpose, he does not have to mask this purpose.

This very simple observation turns Trump's tweets into a litmus test for our algorithms. Provided that our hypothesis is true, we should be able to identify Trump's writing as genuine. Word and word pair distribution anomalies of the kind described above are highly unlikely to turn up in presidential discourse.

## Results and discussion

The distributions of $q$ values calculated for 200 congress and 200 troll tweets of varying lengths (from two to nine content words) and averaged over 30 different random samplings are plotted against each other in Fig 4. Several important things are to be noted here. First, after a certain 'burn-in' period, lines designating tweets that contain different numbers of content words converge towards the asymptotic distribution as the number of tweets in the sample increases.

Second, as we predicted, trolls' tweets of all lengths are characterised by greater values of $q$ than their genuine counterparts. Again, after some period of volatility and uncertainty, the difference in $q$ values tends to be constant as the number of tweets in the sample increases. This is evident from Fig 5, where the mean $q$ values obtained for congress and troll tweets of all possible lengths are plotted.

Third, 50 tweets in the sample seem to be the earliest cut-off point at which genuine and troll messages, regardless of their length, can be identified with high degree of precision. This is the result obtained by using the aforementioned algorithm of establishing the optimal number of tweets needed to detect troll writing. This is also what visual inspection of Fig 6, where the variances of $q$ values in congress and troll tweets of different lengths are plotted, suggests. Starting from this point, we can be confident that the number of content words per tweet in the sample will not affect our results.

This visual impression can be confirmed by logistic regression modelling. We fitted eight binary logistic regression models to the data in order to analyse how the length of a tweet, measured in content words, affected its classifiability. The results are presented in Table 1, which

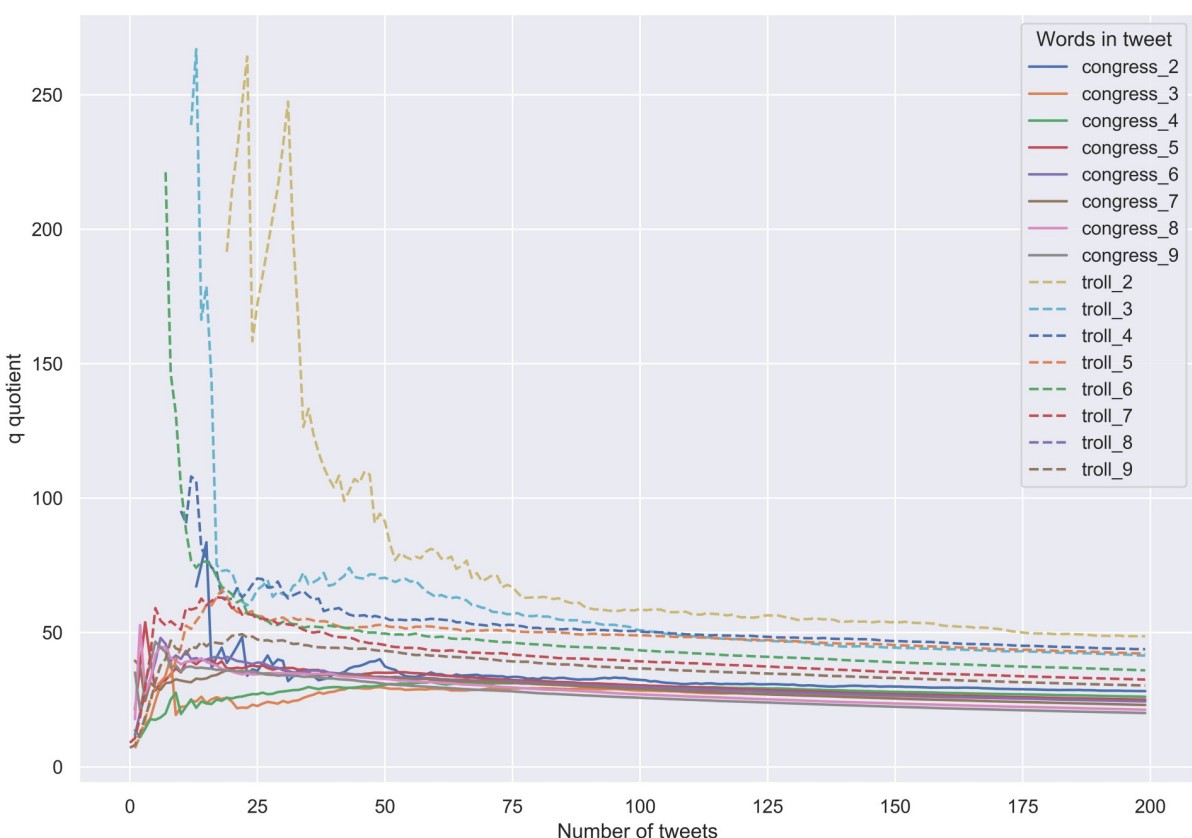

**Fig 4. Congress and troll tweets.** Distribution of $q$.

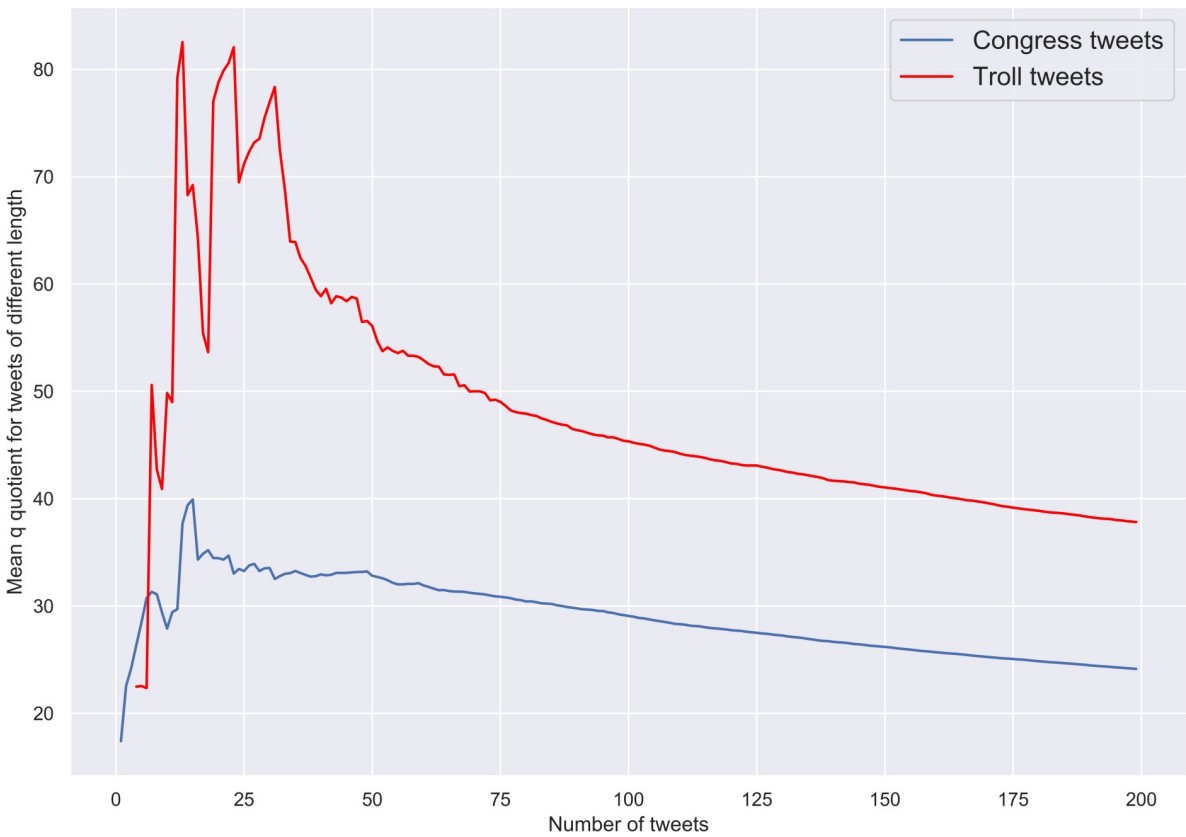

**Fig 5. Congress and troll tweets.** Mean *q* values.

shows, first, that all models were highly significant and, second, that all coefficients were positive. This signifies that with every increase in *q* value the odds of troll writing versus genuine writing become, on average, 1.2 times higher (genuine type was our reference level). However, judging by the log-likelihoods and pseudo *R*-squared numbers, the *q* values appear most useful for the tweets of short to moderate length, that is, containing 3–5 content words.

Taking these coefficients, we can assess the first algorithm for checking whether a certain group of tweets is written by trolls. This algorithm is based on Bayesian inference, in which Bayes' theorem is used to update the probability for a hypothesis as new data become available:

$$P(troll|q) = \frac{P(q|troll) \times P(troll)}{P(q)}$$

We randomly sampled a group of 50 tweets and starting from the first, moved up to the 50th, adding one tweet at a time and updating the probabilities. We assumed a prior probability *P(troll)* to be .5 and calculated the conditional probability *P(q|troll)* by obtaining *q* value for each group of tweets and plugging it into the logistic regression equation. At each subsequent step, the prior probability was substituted with a posterior probability *P(troll|q)* calculated at the previous step. The results for 50 random tweets written by congresspeople, trolls, and Trump were averaged over 100 different samplings (Fig 7).

The figures show that congress tweets are, as we anticipated, aligned with Trump's tweets. However, it is clear that, while troll tweets display a uniform growth of probability of being identified as troll-like as more data are processed, congress and Trump's tweets reveal more

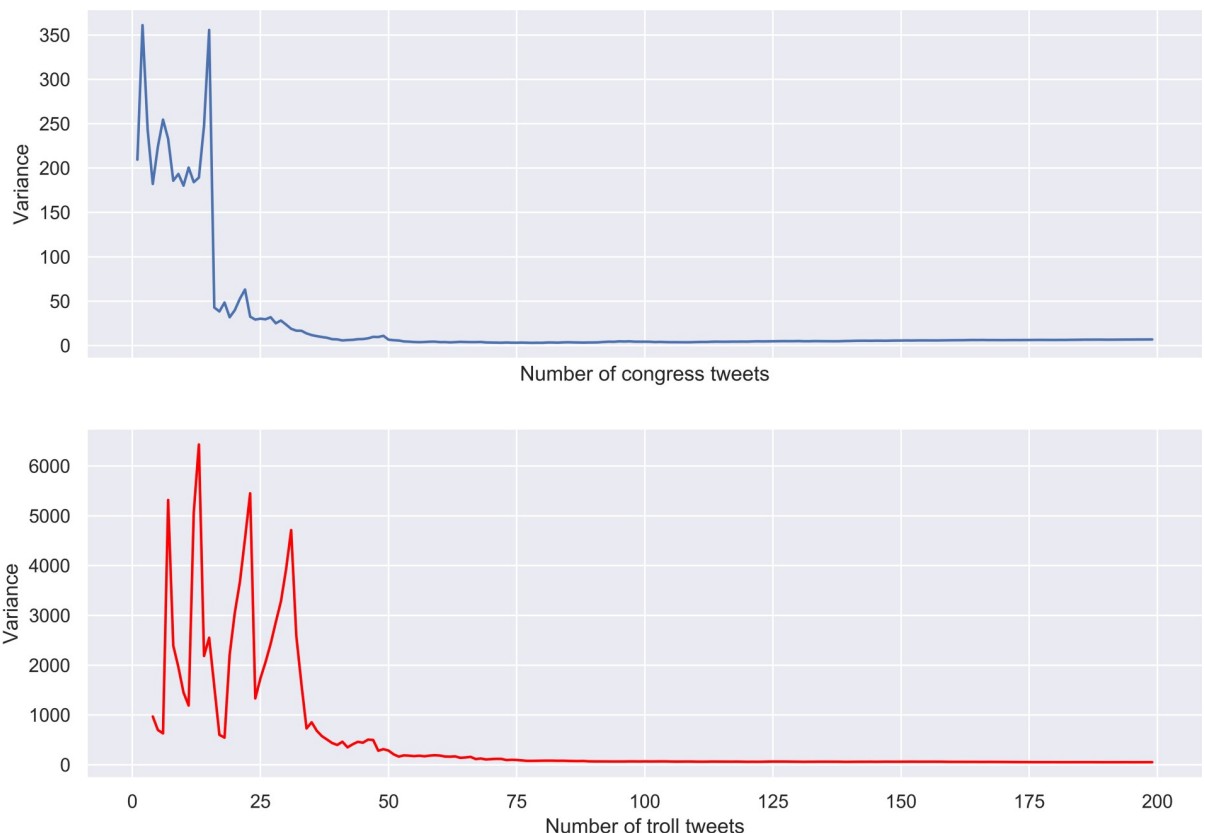

**Fig 6. Congress and troll tweets.** Variance of $q$ values.

complex behaviour. Their respective probabilities plateau and then decline only after approximately 20 tweets are taken into account.

Therefore, it seems reasonable to update posterior probabilities starting not from the first but from the 21st tweet. It will have no bearing on troll tweets and should improve predictions about genuine tweets. We repeated the aforementioned procedure with this correction, the results plotted in Fig 8 support out thesis. The more tweets we add to the initial group of 20, the lower is the posterior probability that these tweets are not genuine. Troll tweets, however, display the reverse trend: the more tweets we add to the initial group of 20, the higher is the posterior probability that we are dealing with troll writing.

Given the random sampling, the results are of course subject to some volatility. For each collection of tweets, we made 100 samples and checked in each of them whether the probability

**Table 1. Logistic regression model's coefficients for tweets of different lengths.**

| Parameters | Number of content words in tweets | | | | | | | |
|---|---|---|---|---|---|---|---|---|
| | **2** | **3** | **4** | **5** | **6** | **7** | **8** | **9** |
| Constant | -3.94 | -10.03 | -8.19 | -9.43 | -6.16 | -4.04 | -6.72 | -4.07 |
| Coefficient for $q$ | 0.09 | 0.3 | 0.2 | 0.24 | 0.16 | 0.11 | 0.2 | 0.13 |
| LL | -213.78 | -86.17 | -130.48 | -116.98 | -205.69 | -234.12 | -198.27 | -232.58 |
| p-value | < 0.001 | < 0.001 | < 0.001 | < 0.001 | < 0.001 | < 0.001 | < 0.001 | < 0.001 |
| Pseudo $R^2$ | 0.22 | 0.68 | 0.52 | 0.57 | 0.25 | 0.15 | 0.28 | 0.16 |

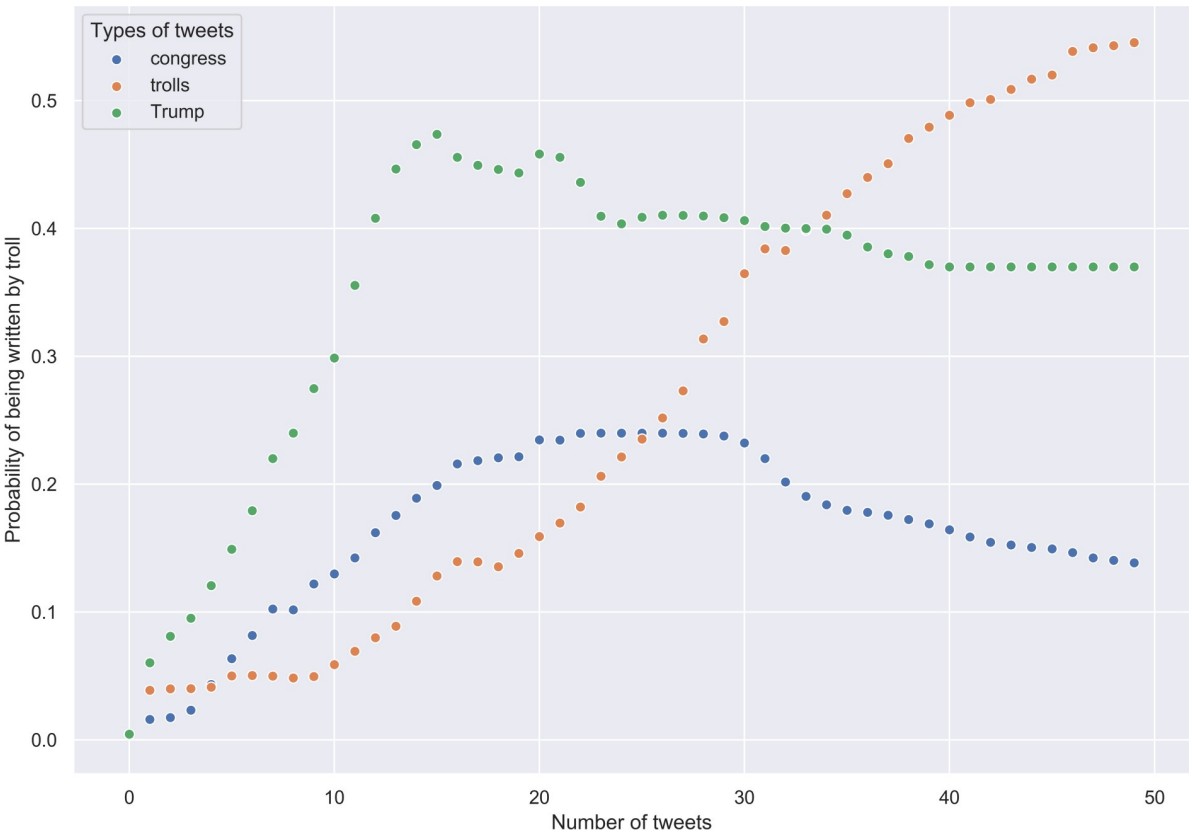

**Fig 7. Congress, troll, and Trump's tweets.** Results of Bayesian inference for 50 random tweets (updating from the first tweet in the sample).

of being classified as troll writing increases or decreases in them from the 21st to the 50th tweet. In case of continuous increase, we assigned the *troll* label to the sample, in case of decrease, the *genuine* label. The procedure was repeated 30 times for all three collections of tweets. The percentages of correct group identification are given in Table 2.

On average, out of 100 samples of 50 tweets each, 86 samples of congress tweets were identified as genuine, 70 samples of troll tweets were identified as troll-like, and 75 samples of Trump's tweets were identified as genuine. We could argue that these far from ideal results are mainly due to the fact that we analyse random samples where tweets are not sequentially (i. e. chronologically) aligned. The question is, however, whether we can achieve higher accuracy with the help of another algorithm.

The second proposed algorithm is based on simple linear regression model coefficients. We fitted separate linear regression models to the congress and troll tweet data to evaluate the response variable of $q$ against two independent variables: 1) the number of tweets in a group and 2) the number of words in a tweet. The coefficients for the congress and troll models are presented in Tables 3 and 4, respectively.

Both models are significant. Interestingly, the coefficients are of opposite signs: with congress tweets, the $q$ values increase as the tweets become longer and the number of tweets in the group becomes larger. Troll tweets, in contrast, reveal a decrease in the $q$ values under the same conditions. In essence, these are manifestations of the same pattern observed earlier with logistic regression. The distinction is best made with relatively small groups of short and medium-length tweets. The larger the numbers, the more genuine and troll writing tend

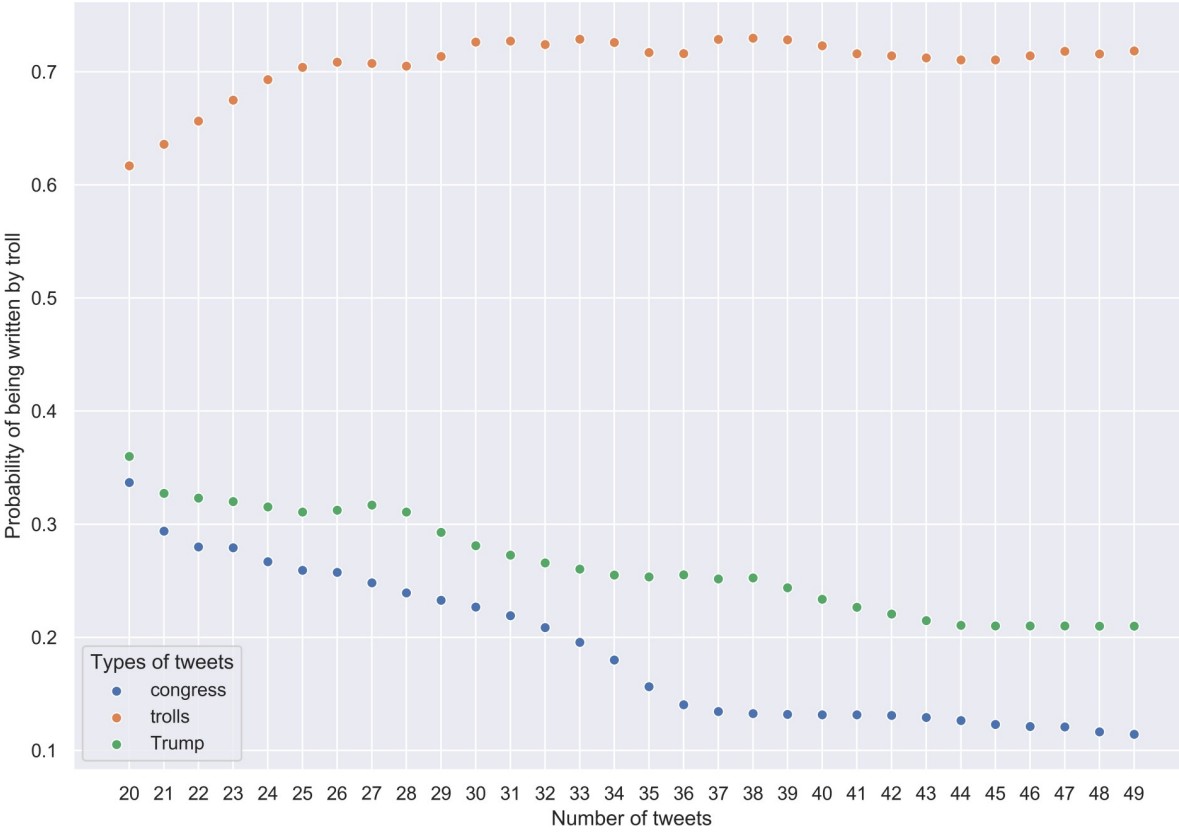

**Fig 8. Congress, troll, and Trump's tweets.** Results of Bayesian inference for 50 random tweets (updating from the 21st tweet in the sample).

towards each other. The negative interaction term in both cases signifies that after some cut-off point this trend gets reversed.

Taking the coefficients of both models, we assessed the second algorithm for checking whether a certain group of tweets is written by trolls. This algorithm works as follows:

1. a group of 50 tweets is taken;

2. a subgroup of 20 tweets is randomly sampled from these 50 tweets;

3. the $q$ value is calculated for the subgroup ($q_s$);

4. two alternative $q$ values for the same subgroup are obtained by plugging the number of tweets and the mean length of tweets into both linear regression equations and predicting the respective values of $q$: one that, given the quantitative parameters of the sample, would be characteristic of genuine writing ($q_c$) and another of troll writing ($q_t$);

**Table 2. Bayesian inference for 50 random tweets (% of correct group identification).**

| Group of tweets | Statistics | | | |
|---|---|---|---|---|
| | **M** | **SD** | **95 CI lower** | **95 CI upper** |
| Congress | 86.3 | 3.45 | 85.05 | 87.54 |
| Trolls | 69.7 | 4.18 | 68.2 | 71.19 |
| Trump | 75.33 | 4.26 | 74 | 77.05 |

**Table 3. Coefficients of linear regression model (congress tweets).**

| | B (SE) | 95% CI | |
| --- | --- | --- | --- |
| | | Lower | Upper |
| Constant | 23.78*** (0.78) | 22.24 | 25.33 |
| Number of words | 2.19*** (0.13) | 1.93 | 2.45 |
| Number of tweets | 0.05*** (0.006) | 0.04 | 0.07 |
| Interaction term | -0.02*** (0.001) | -0.02 | -0.01 |

Adjusted $R^2$ = 0.36, $p < .0001$. Significance codes:

*** $p < 0.001$.

5. two squared distances $(q_c—q_s)^2$ and $(q_t—q_s)^2$ are calculated; if $q_0$ is closer to $q_c$, the subgroup is classified as genuine; if it is closer to $q_t$, then the subgroup is classified as troll-like;

6. steps 2–5 are repeated 100 times and the prevalent decision is pronounced as a final verdict. (We made the algorithm more biased in favour of genuine writing: troll decisions should outnumber genuine ones by at least one fourth for the sample to be labelled as troll writing.)

To assess the accuracy of the second algorithm, we checked 100 samples from all three collections of tweets. The procedure was repeated 30 times. The percentages of correct group identification are given in Table 5.

On average, out of 100 samples of 50 tweets each, 90 samples of congress tweets were identified as genuine, 90 samples of troll tweets were identified as troll-like, and 88 samples of Trump's tweets were identified as genuine.

Combining these two algorithms and using the results of the frequentist one as prior probabilities for the Bayesian one leads to even better and more consistent results. Thus, 95% confidence intervals show that the true mean of the numbers of correctly identified samples out of 100 samples lies between 89 and 90 for congresspeople; between 91 and 93 for trolls; and between 85 and 88 for Trump.

The sensitivity and specificity of our model can be estimated as follows (Table 6):

Two more things should be mentioned. First, there may be concern about how representative our results are given the small sample size and whether the same results hold if the analysis is done on the whole dataset. The fact is they do not only hold but improve as we increase the sample size. Thus, for the samples of 100, 250, and 1000 tweets we got the following numbers (Table 7):

In other words, the more tweets we have at our disposal, the more accurate prediction about their troll or genuine nature we can make. Our main task, however, was to identify the

**Table 4. Coefficients of linear regression model (troll tweets).**

| | B (SE) | 95% CI | |
| --- | --- | --- | --- |
| | | Lower | Upper |
| Constant | 62.72*** (2.12) | 58.55 | 66.89 |
| Number of words | -1.32*** (0.35) | -2.02 | -0.62 |
| Number of tweets | -0.04* (0.01) | -0.07 | -0.006 |
| Interaction term | -0.009** (0.003) | -0.01 | -0.003 |

Adjusted $R^2$ = 0.18, $p < .0001$. Significance codes:

*** $p < 0.001$

** $p < 0.01$

* $p < 0.05$.

**Table 5. Frequentist inference for 50 random tweets (% of correct group identification).**

| Group of tweets | Statistics | | | |
|---|---|---|---|---|
| | **M** | **SD** | **95 CI lower** | **95 CI upper** |
| Congress | 89.8 | 3.1 | 88.68 | 90.91 |
| Trolls | 90.17 | 3.14 | 89.05 | 91.3 |
| Trump | 88.06 | 3.43 | 86.83 | 89.29 |

**Table 6. Sensitivity and specificity for the samples of 50 tweets.**

| | **Troll tweets (%)** | **Genuine tweets (%)** |
|---|---|---|
| Troll test positive | 92 | 12 |
| Troll test negative | 8 | 88 |

**Table 7. Sensitivity and specificity for the samples of 100, 250, and 1000 tweets.**

| | **Troll tweets (%)** | **Genuine tweets (%)** |
|---|---|---|
| 100 tweets | | |
| Troll test positive | 98 | 5 |
| Troll test negative | 2 | 95 |
| 250 tweets | | |
| Troll test positive | 100 | 2 |
| Troll test negative | 0 | 98 |
| 1000 tweets | | |
| Troll test positive | 100 | 1 |
| Troll test negative | 0 | 99 |

earliest possible cut-off point, so that the dissemination of propaganda could be stopped as soon as possible. The provided analysis shows that with 50 tweets, classification is fairly accurate, with 1000 tweets, it is close to impeccable.

Another question is whether the proposed method can be employed to detect tweets written by state-sponsored trolls in languages other than English. Given the pure semiotic, non-language specific nature of this method, we expect to find that it will work with other languages as well. To substantiate this claim, we extracted 388,688 tweets written in Russian from the same collection of Russian trolls' tweets posted on Github by Linvill and Warren.

The same procedure as described above was applied to these tweets. Out of 100 random samples of 50 tweets each, 78 samples were identified as troll-like. Out of 100 random samples of 100 tweets each, 93 samples were identified as troll-like. Increasing sample size up to 250 tweets resulted in the accuracy of 98% and with 1000 tweets, we got 99% of correct predictions. These results suggest that, though further research is needed to analyse different tactics employed by different types of state-sponsored trolls, we have good reasons to believe that correlation of the numbers of repeated content words and content word pairs is different in genuine and troll writing, regardless of language.

## Conclusion

In this paper, we developed a special quantitative measure for identifying troll tweets which does not depend on specific communicative situations and is, presumably, applicable not only

to the messages written in English. Though troll writing is usually thought of as being permeated with recurrent messages, we showed that its most characteristic trait is anomalous distribution of repeated words and word pairs.

The origins of this trait can be explained by the sociolinguistic restrictions of troll writing, namely by a combination of two dominant factors: a necessity to both communicate a limited number of messages multiple times and to fake the diversity of contexts and topics in the process. One could argue that in order to find more words to express the same concept, trolls may also extensively use formal jargon, otherwise uncommon in digital communication, as Lundberg and Laitinen suggest.

Using the quantitative measure of the ratio between proportions of repeated words and word pairs, we showed that only 50 tweets are needed to make reliable statements about their true nature. We tested two algorithms for identifying genuine and troll tweets, one employing frequentist inference and another employing Bayesian inference, and showed that the most consistent results are achieved by combining these algorithms.

As a litmus test for our method, we used Donald Trump's tweets. Given that we were able to identify Trump's writing as genuine, despite his provocative and, according to Twitter's new policy of labelling posts, 'potentially misleading' manner of tweeting, our approach can be considered well-founded. Trump cannot be called a troll in our sense of the word, because his communicative behaviour is not a combination of the two aforementioned dominant factors. While he undoubtedly repeats himself over and over again, he does not use different topics as simple proxies for getting through a very limited number of signals. Our findings accord with those of Clarke and Grieve [25], who pointed out that the style of Trump's tweets varied systematically over time depending on his communicative goals.

It is important to mention that the proposed method was tested, apart from troll messages, only against tweets of public figures (congresspeople and Trump). We still need to investigate whether the results will hold with other types of messages.

## Supporting information

**S1 Fig. Congress tweets.** Circos plot of a network of words (nodes) and pairs of words (edges).
(TIF)

**S2 Fig. Troll tweets.** Circos plot of a network of words (nodes) and pairs of words (edges).
(TIF)

## Acknowledgments

I would like to express my deepest love and gratitude to Igor Monakhov, my father, for his patient help and thoughtful advice.

## Author Contributions

**Investigation:** Sergei Monakhov.

**Methodology:** Sergei Monakhov.

**Writing – original draft:** Sergei Monakhov.

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
