## [Decision Letter · Decision Letter 0]

5 Jun 2020

PONE-D-20-11657

How many tweets are needed to identify an internet troll?

PLOS ONE

Dear Dr. Monakhov,

Thank you for submitting your manuscript to PLOS ONE. After careful consideration, we feel that it has merit but does not fully meet PLOS ONE’s publication criteria as it currently stands. Therefore, we invite you to submit a revised version of the manuscript that addresses the points raised during the review process.

In particular, please address the concerns of Reviewer 2 about the small size of the sample and the generalization of the results.

Please also clarify the difference between trolls and social bots, the details of the tweet sampling and address the other concerns of Reviewer 1.

We look forward to receiving your revised manuscript.

Kind regards,

Alexandre Bovet, Ph.D.

Academic Editor

PLOS ONE

Journal Requirements:

2. Please amend your title to comply with PLOS publicaiton guidelines: https://journals.plos.org/plosone/s/submission-guidelines#loc-title In this case the title is not sufficiently descriptive of the conducted research.

Additional Editor Comments (if provided):

Reviewers' comments:

Reviewer's Responses to Questions

**Comments to the Author**

1. Is the manuscript technically sound, and do the data support the conclusions?

Reviewer #1: Yes

Reviewer #2: Partly

2. Has the statistical analysis been performed appropriately and rigorously? 

Reviewer #1: Yes

Reviewer #2: Yes

3. Have the authors made all data underlying the findings in their manuscript fully available?

Reviewer #1: Yes

Reviewer #2: Yes

4. Is the manuscript presented in an intelligible fashion and written in standard English?

Reviewer #1: Yes

Reviewer #2: Yes

5. Review Comments to the Author

Reviewer #1: This work introduces a linguistic analysis of word co-occurrences and repetitions in social discourse promoted by normal users and internet trolls. The author introduces a simple methodology based on a small number of tweets and leading to a classifier of normal/troll posts with high accuracy.

The manuscript is well written, introducing the reader to the relevant topic of understanding the malicious phenomenon of internet trolling during massive online events and detecting trolls in online platforms. The methodology is simple but novel and results are interesting and well explained. The author should be praised for the innovative combination of simple machine learning techniques, Bayesian inference and sociolinguistic analysis.

There are only a few points, mainly a few missing references and some minor adjustments, that once addressed would improve the quality of this manuscript.

For all the above reasons, I recommend publication of this manuscript in PLoS One with minor revisions.

My points are down below:

• In the Introduction, when the author mentions previous linguistic investigations of trolls’ language, the recent work by Lundberg and Laitinen (2020) might be referenced and briefly discussed. The authors investigated 3.5 million tweets from trolls’ social discourse on Twitter in 2018 but they did not follow a machine learning classification approach as outlined in the current manuscript. Through a simpler frequency analysis, Lundberg and Laitinen also showed that trolls use a smaller linguistic vocabulary, with a sometimes more formal tone. Their results would nicely complement the independent finding presented in this work, on a different dataset and with a different methodology, and could also be interpreted under the light that in order to find more words to express the same concept, not only do trolls mix different contexts but also end up using formal jargon uncommonly used in current language.

• In the Introduction, it is not only trolls that represent a powerful weapon for shifting public opinion but also social bots, fake accounts entirely managed by a software. Do social bots fall into the author’s classification of internet trolls? A recent work by Stella et al. (2019) identified a category of “augmented” human Twitter users that achieve trolling by acting normal but then having their messages re-tweeted or favourited by a large number of social bots.

• In the Results, Figures 1 and 2 are based on sampling 1000 random tweets. How many of these samplings were performed?

• Figures 3 and 4 seem very difficult to parse. Maybe they could be moved to an Appendix. Also, were these networks based on tweets with the same length? The hierarchical edge bundling visualisation seems to suggest that trolls’ linguistic network is more sparse than non-trolls’ but other than that it is of little use here.

• Figure 5-8 seem quite noisy in their left parts. Have they been averaged over 30 different random samplings? As they stand right now, it is really hard to follow the claims from the main text (e.g. two lines “stop mingling after 20 tweets”). Please provide a better labelling, consider changing the scale, smoothing the curves with additional numerical averaging or inserting some insets to highlight specific areas.

• When reporting the accuracy of the classification, could anything be said about what types of errors does the classification perform, e.g. does it tend to identify “false” trolls or “false” normal users?

• About using Trump’s tweets as a reference corpus for the algorithms, it should be noted that a recent work by Clarke and Grieve (2019) pointed out how Trump’s tweets continuously shift between different linguistic paradigms, which would explain why his language does not have high repetition counts like the ones detected in trolls here.

• Maybe “abusive” is not the correct wording for defining Trump’s tweets in a scientific publication without a supporting reference. It is only rather recently that a third party, in this case Twitter, started analysing the semantic content of Trump’s tweets and identified some of them as “potentially misleading” (CNN, 2020). Maybe the last few sentences of the Discussion might be reworded to accommodate specific references outlining how the POTUS’s tweets are controversial, biased and enflaming.

• Typos:

• Line 60 – epy widely?

Reviewer #2: This is an interesting paper that aims to study the differences between tweets posted by congress and state-sponsored trolls.

However, i have serious concerns about the generalizability of the proposed methods.

First, the paper's analysis and results are based on a very small sample of the whole dataset (like 200 or 1000 at some cases tweets out of 1M tweets). This raises concerns on how representative the results are and whether the same results hold if the analysis is done on the whole dataset. I suggest to the authors to expand their analysis so that it includes all the tweets from their dataset rather than sampling a very small percentage of the tweets.

Second, the authors claim that their approach can be used for other languages as well. I think, since each language its different and each type of state-sponsored trolls might employ different tactics, this is a somewhat unsubstantiated claim. I suggest to the authors to expand their analysis to include data from other languages and types of state-sponsored trolls so that we can better assess the generalizability of the proposed methods.

6. PLOS authors have the option to publish the peer review history of their article (what does this mean?). If published, this will include your full peer review and any attached files.

Reviewer #1: No

Reviewer #2: No

---

## [Author Response · Author response to Decision Letter 0]

19 Jun 2020

Dear editors and reviewers, please accept our sincere gratitude for all your comments and suggestions. They helped to improve the paper a lot! As we stated in the attached Response to Reviewers file, we carefully tried to address all the issues raised in your reviews.

---

## [Decision Letter · Decision Letter 1]

30 Jun 2020

PONE-D-20-11657R1

Early detection of internet trolls: Introducing an algorithm based on word pairs / single words multiple repetition ratio

PLOS ONE

Dear Dr. Monakhov,

Thank you for submitting your manuscript to PLOS ONE. After careful consideration, we feel that it has merit but does not fully meet PLOS ONE’s publication criteria as it currently stands. Therefore, we invite you to submit a revised version of the manuscript that addresses the points raised during the review process.

We thank you for the revisions that both reviewers appreciated. Before acceptance, we ask you to address the last issue raised by Reviewer 1. Namely, replicating the analysis of Table 7 for 100, 250 tweets and, if possible, 1000 tweets.

We look forward to receiving your revised manuscript.

Kind regards,

Alexandre Bovet, Ph.D.

Academic Editor

PLOS ONE

Reviewers' comments:

Reviewer's Responses to Questions

**Comments to the Author**

1. If the authors have adequately addressed your comments raised in a previous round of review and you feel that this manuscript is now acceptable for publication, you may indicate that here to bypass the “Comments to the Author” section, enter your conflict of interest statement in the “Confidential to Editor” section, and submit your "Accept" recommendation.

Reviewer #1: All comments have been addressed

Reviewer #2: All comments have been addressed

2. Is the manuscript technically sound, and do the data support the conclusions?

Reviewer #1: Yes

Reviewer #2: Yes

3. Has the statistical analysis been performed appropriately and rigorously? 

Reviewer #1: Yes

Reviewer #2: Yes

4. Have the authors made all data underlying the findings in their manuscript fully available?

Reviewer #1: Yes

Reviewer #2: Yes

5. Is the manuscript presented in an intelligible fashion and written in standard English?

Reviewer #1: Yes

Reviewer #2: Yes

6. Review Comments to the Author

Reviewer #1: The author took great care in addressing all my comments and I am positively impressed with the final quality of the manuscript.

I recommend acceptance of the manuscript. I only have one final suggestion about line 424 "First, an anonymous reviewer expressed concern about how representative our results are given the small sample size...". I am confident that any reader could get the same doubt so the reference to the specific reviewer here could be changed. Also, in order to further dissipate these doubts and the risk of overfitting, I would try to replicate the analysis on the new Table 7 for 100 tweets (which is still a relatively small number) also in case of 250 tweets (which becomes more of a diary/storyline) and, if possible, 1000 tweets.

Consider that results cannot be monotonously impeccable as the higher the number of tweets used in the analysis the larger the linguistic scope or set of topics identified in the analysis and thus, because of such larger variability, the accuracy of the model could actually decrease rather than increase. I think this is an important reality check in order to either support or change the authors' claim about the model being "impeccable". If the model kept performing very well also for larger sizes of tweets, this would mean the important possibility of using this methodology not only on Twitter but also on platforms with longer texts like Facebook, a possibility that would be advantageous for the author and his very interesting work.

It is also remarkable that the methodology works also cross-linguistically in Russian, as shown by the author.

I will let the Editor decide on this final point, whether to accept the paper as is or request this minor adjustment. I repeat I was satisfied with the author's work.

Reviewer #2: I would like to thank the authors for addressing the reviewers' comments and resubmitting their work. All the reviewers' comments were addressed in the revised manuscript and the manuscript is substantially stronger now. I recommend acceptance of this work.

7. PLOS authors have the option to publish the peer review history of their article (what does this mean?). If published, this will include your full peer review and any attached files.

Reviewer #1: No

Reviewer #2: No

---

## [Author Response · Author response to Decision Letter 1]

8 Jul 2020

1) The reference to the specific reviewer has been changed. 

2) The analysis has been replicated for the sample sizes of 100, 250, and 1000 tweets.

---

## [Editor Report · Decision Letter 2]

15 Jul 2020

Early detection of internet trolls: Introducing an algorithm based on word pairs / single words multiple repetition ratio

PONE-D-20-11657R2

Dear Dr. Monakhov,

We’re pleased to inform you that your manuscript has been judged scientifically suitable for publication and will be formally accepted for publication once it meets all outstanding technical requirements.

Kind regards,

Alexandre Bovet, Ph.D.

Academic Editor

PLOS ONE

Additional Editor Comments:

When preparing the final version, please take into account the following comments:

- Please mention the limitations of your method in the introduction and conclusion. In particular that you only tested you method against tweets of public figures (congresspeople and Trump).

- Please define what is a clique (line 177) in graph theory.

- Rename "Appendix" to "Supporting information" (see https://journals.plos.org/plosone/s/submission-guidelines#loc-manuscript-organization)

---

## [Editor Report · Acceptance letter]

17 Jul 2020

PONE-D-20-11657R2 

Early detection of internet trolls: Introducing an algorithm based on word pairs / single words multiple repetition ratio 

Dear Dr. Monakhov:

I'm pleased to inform you that your manuscript has been deemed suitable for publication in PLOS ONE. Congratulations! Your manuscript is now with our production department. 

Kind regards, 

on behalf of

Dr. Alexandre Bovet 

Academic Editor

PLOS ONE